# High Levels of ROS Impair Lysosomal Acidity and Autophagy Flux in Glucose-Deprived Fibroblasts by Activating ATM and Erk Pathways

**DOI:** 10.3390/biom10050761

**Published:** 2020-05-13

**Authors:** Seon Beom Song, Eun Seong Hwang

**Affiliations:** Department of Life Science, University of Seoul, Dongdaemun-gu, Seoulsiripdae-ro 163, Seoul 02504, Korea; scitiger@never.com

**Keywords:** ATP, glucose deprivation, autophagy, autophagy flux, ataxia-telangiectasia mutated (ATM), reactive oxygen species (ROS), extracellular-signal-regulated kinase (Erk)

## Abstract

Under glucose deprivation, cells heavily mobilize oxidative phosphorylation to maintain energy homeostasis. This leads to the generation of high levels of ATP, as well as reactive oxygen species (ROS), from mitochondria. In nutrient starvation, autophagy is activated, likely to facilitate resource recycling, but recent studies suggest that autophagy flux is inhibited in cells undergoing glucose deprivation. In this study, we analyzed the status of autophagic flux in glucose-deprived human fibroblasts. Although lysosomes increased in quantity due in part to an increase of biogenesis, a large population of them suffered low acidity in the glucose-deprived cells. Autophagosomes also accumulated due to poor autolysis in these cells. A treatment of antioxidants not only restored lysosomal acidity but also released the flux blockade. The inhibition of ataxia telangiectasia mutated (ATM) serine/threonine kinase, which is activated by ROS, also attenuated the impairment of lysosomal acidity and autophagic flux, suggesting an effect of ROS that might be mediated through ATM activation. In addition, the activity of extracellular signal-regulated kinase (Erk) increased upon glucose deprivation, but this was also compromised by a treatment of antioxidants. Furthermore, the Erk inhibitor treatment also alleviated the failure in lysosomal acidity and autophagic flux. These together indicate that, upon glucose deprivation, cells undergo a failure of autophagy flux through an impairment of lysosomal acidity and that a high-level ROS-induced activation of Erk and ATM is involved in this impairment.

## 1. Introduction

Autophagy is an important cellular salvage mechanism of recycling resources for anabolic building blocks, as well as energy. Autophagy also functions as a process of damage disposition and thereby helps cells maintain homeostasis and control cell fate in aging and carcinogenesis as well [1]. Studies on glucose deprivation, a widely studied model of nutrient deprivation where cells are believed to suffer low level energy resources, have shown that the activation of AMP-activated protein kinase (AMPK), a well-conserved energy-sensing kinase, is a major molecular event that triggers autophagy activation [2]. Activated AMPK phosphorylates and activates UNC51-like kinase (ULK1), an inducer of autophagy [3], and key upstream regulators of autophagosome formation [4,5,6,7,8]. Therefore, it has been believed that, in conditions of low glucose, a decrease in ATP level would activate AMPK and, thereby, drive autophagy-mediated resource recycling [9,10,11,12,13,14]. 

However, recent studies demonstrated that, in glucose-deprived cells, fatty acid β-oxidation and mitochondrial oxidative phosphorylation (OXPHOS) are increased [9,10,11,12,13,14,15]. These changes would help cells maintain energy homeostasis in the absence of glucose input [11,16], but the activation of a high level of OXPHOS places cells under increased oxidative stress [10,11,12,13,15]. We reported an increase of the levels of reactive oxygen species (ROS) along with an elevation of ATP generation in glucose-deprived human normal fibroblasts, as well as tested cancer cells [17]. High-rate OXPHOS elevates superoxide radical generation, likely through an electron leak from the transport chain [18]. An increase in the mitochondria contents, which also takes place in glucose-deprived cells [10], would amplify ROS generation, along with an increase in ATP synthesis. Mitochondria-generated ROS are known to control cellular signal transduction and play roles in cell homeostasis [19]. However, when produced at high levels, they appear to intervene with normal cell physiology in part by altering the activities of certain factors such as AMPK. In fact, evidence indicates that AMPK is activated in glucose-deprived cells by an increase of ROS [17,20]. This appears to make sense in that, in the presence of high-level ROS and, thereby, the heavy accumulation of oxidatively damaged adducts, cells need to accelerate autophagy to facilitate damage disposition. Accordingly, our previous work found a substantial elevation of autophagosome generation in glucose-deprived cells [17]. However, we suspected that this high level of autophagosome accumulation might be attributed to poor autophagy flux. Consistent with this idea, an impairment of the flux of basal autophagy and of those induced by other stimuli upon glucose deprivation has been reported [21]. 

Autophagy proceeds largely in two phases: sequestration of cargo materials in double-membrane structures (autophagosome formation) and the formation of autolysosomes and the subsequent degradation of materials (autolysis). So far, research has intensively investigated the cellular and molecular mechanisms underlying autophagy initiation [22]. However, how autophagy flux is regulated is poorly understood. Furthermore, although the phenomenon of an autophagy flux blockade has been reported in recent studies, the nature and causes of the impaired process are not well-known. Understanding these processes bears importance, since certain pathological conditions are suspected to involve impaired autophagy flux. For example, neurodegenerative diseases, such as Alzheimer’s, Parkinson’s, and Huntington’s diseases, are associated with the accumulation of massive autophagic vacuoles [23]. These findings have suggested that the accumulation of autophagic vacuoles or autophagy flux impairment is a causative event of these diseases [24]. Furthermore, autophagy flux has been suggested to play essential roles in the survival of cardiac myocytes under ischemia or oxidative stress [25]. Therefore, understanding how the impairment of autophagy flux is driven in cells and how it interferes with cellular or tissue functions not only furthers our understanding of the in vivo function of autophagy but also has great clinical importance.

In the current study, we examined autophagy flux in glucose-deprived human fibroblasts in an attempt to determine the mechanisms underlying this impairment of autophagy flux. Autolysosome formation (autophagosome-lysosome fusion) and autolysosomal degradation are the two key steps in the flux of autophagy [26]. For the latter, lysosomal function is critical. Recent reports suggest an involvement of an impairment of vacuolar ATPase (vATPase), a proton pump in acidic compartments that include lysosomes, in lysosomal dysfunction associated with various pathologies and aging as well [27]. One of the factors that affect the activity of vATPase is ataxia telangiectasia mutated (ATM), a key mediator of DNA damage signaling and repair [28]. It has been shown that ATM is activated by mitochondria-generated ROS to increase the cellular antioxidant capacity [29]. The inhibition of ATM activation was shown to help maintain vATPase activity, as well as lysosome function, in senescent cells [30]. Meanwhile, ROS also activate a mitogen-activated signaling cascade [31]. Among them, extracellular signal-regulated kinase (Erk) is a recently suggested target of activation by ROS generated from mitochondria [32,33]. Active Erk has also been shown to attenuate autolysis through decreasing the cathepsin protein levels [34]. 

In this study, we examined the status of autolysosomes in glucose-deprived fibroblasts and found that multiple signaling pathways that involve ATM and Erk are triggered by high levels of ROS to impair lysosome acidity and cause autophagy flux failure upon glucose deprivation. Our results also propose another example of dysfunction in cellular activity that is caused by high-level ROS generation.

## 2. Materials and Methods

### 2.1. Cell Culture and Chemical Treatments

Normal human fibroblasts isolated from healthy newborn foreskins and provided by Dr. Jin-Ho Chung (IRB No. H-1101-116-353 of the School of Medicine, Seoul National University, Korea) were maintained in Dulbecco’s modified Eagle’s medium (DMEM; LM001-11, Welgene, Daegu, Korea) supplemented with 10% fetal bovine serum (S-FBS-US-015, Serana, Bunbury, Australia) at 5% CO_2_ and 37 °C. DMEM with no glucose (LM001-79, Welgene) was used for glucose deprivation. In the refeeding experiments, glucose was added to the glucose-free medium to a 5.5-mM concentration. Chemicals listed in the following table were applied to cells for the indicated effects (Table 1)

### 2.2. Western Blotting Analysis

Proteins extracted in RIPA buffer (50 mM Tris-HCL (pH 7.5), 150 mM NaCl, 1% Nonidet P-40, 0.5% sodium deoxycholate, 0.1% sodium dodecyl sulfate (SDS)) supplemented with NaF, NaVO_4_, and a protease inhibitor cocktail (Sigma-Aldrich, P2714) were separated by SDS-polyacrylamide gel electrophoresis and transferred to a nitrocellulose membrane. Membranes were blotted with antibodies against human LC3 (#2775); phospho-AMPKα (Thr172, #2535); AMPKα (#2603) (from Cell Signaling Technology, Beverly, USA); Erk (SC-93); phospho-ERK (Tyr204 and SC-7383); ERK1 (SC-376852); p62 sequestosome (SC-25575); Lamp1 (SC-20011); Lamp2 (SC-5571); Limp2 (SC-55570) (from Santa Cruz Biotechnology, Dallas, TX, USA); and β-actin (A5441, Sigma-Aldrich). Membranes were then incubated with horseradish peroxidase-conjugated secondary antibodies, and protein bands were visualized using SuperSignal West Femto substrate (Thermo Fisher, Waltham, MA, USA).

### 2.3. Immunofluorescence and Confocal Microscopy

Cells grown on coverslips were fixed in 3.7% paraformaldehyde in PBS for 20 min, permeabilized with 0.1% Triton X-100 in phosphate-buffered saline (PBS) for 15 min, blocked with 10% FBS in PBS for 2 h, and incubated with primary antibodies overnight. For the detection of autophagosomes or lysosomes, antibodies against LC3 or Lamp1, respectively, were used. Cells were washed and incubated with Alexa Fluor 488-conjugated anti-rabbit, Alexa Fluor 633-conjugated anti-mouse, Alexa Fluor 488-conjugated anti-mouse, Alexa Fluor 405-conjugated anti-mouse, Alexa Fluor 546-conjugated anti-rabbit, or Alexa Fluor 546-conjugated anti-mouse secondary antibodies (all from Thermo Fisher) for 2 h and visualized under a confocal microscope (LSM 510, Carl Zeiss, Thornwood, NY, USA). Numbers of puncta were counted using ImageJ analysis software (NIH, Bethesda, MA, USA).

### 2.4. Transfection of DNA or siRNA 

Cells were transfected with a plasmid expressing RFP-GFP tandem-tagged LC3 protein (ptfLC3 [40], #21074, Addgene, Cambridge, MA, USA) using Lipofectamine^®^ 2000 transfection reagent (11668-027, Invitrogen, Carlsbad, CA, USA) according to the manufacturer’s protocol. Cells were transfected with small interfering RNA (siRNA) targeting AMPKα1 (siAMPK) or control siRNA (siCtrl) using lipofectamine RNAiMAX (13778-150, Thermo Scientific, Waltham, MA, USA) according to the manufacturer’s protocol. siAMPK (5562-1) and control RNA (siCtrl; CCUACGCCACCAAUUUCGU (dTdT)) were purchased from Bioneer, Daejun, Korea.

### 2.5. Flow Cytometry for Determination of Autophagic Flux 

The CYTO-ID^®^ autophagy detection kit (ENZ-51031-0050, Enzo Life Science, Farmingdale, USA) was used to monitor autophagic flux in live cells according to the manufacturer’s protocol. The same number of fibroblasts were seeded on 60-mm dishes. Approximately 10 h after seeding, cells were further incubated for 72 h regardless of a difference in treatment and then collected for CytoID analysis. For control, cells were incubated in a glucose-containing DME medium throughout three days. For 2-day starvation, cells were incubated in a glucose-containing medium for 1 day, and then, in glucose-free DMEM for two days. For 3-day starvation, cells were incubated in glucose-free DMEM for three days. At the end of the three-day incubation, cells were trypsinized and stained with CYTO-ID. Cells were then washed and analyzed by flow cytometry using FACS Canto II (BD Biosciences).

### 2.6. Quantitative Polymerase Chain Reaction (qPCR)

Total RNA was isolated using TRIzol reagent (total RNA isolation solution, Thermo Fisher) and converted to cDNA using a reverse transcription kit (ReverTra Ace qPCR RT kit, FSQ-101, TOYOBO, Osaka, Japan). One microgram of cDNA diluted in water was analyzed by qPCR using SYBR Green on a CFX connect (BioRad, Hercules, USA). The following PCR primers were used: 5′-CCGTACTCCATCCCTCCCTG-3′ and 5′-GTAGGTCGGGCTGTAGCCAG-3′ (cathepsin B), 5′-TGGGCGGTGTCAAAGTGGAG-3′ and 5′-GCCCAGGATGCCATCGAACT-3′ (cathepsin D), 5′-ACCACCGTCCTGCTCTTCCA-3′ and 5′-AGGGTCTCTGGCGTCAGGAA-3′ (Lamp1), 5′-CACACCACTGTGCCATCTCCT-3′ and 5′-CCCATGGTAGCCAGCAGACA-3′ (Lamp2), and 5′-TGGCATTGCCGACAGGAT-3′ and 5′-GCATTTGCGGTGGACGAT-3′ (β-actin).

### 2.7. Measurement of Lysosomal Aacidity and Lysosomal Enzyme Activity 

Lysosensor yellow/blue DND-160 (Thermo Fisher, L7545) exhibits both dual-excitation and dual-emission spectral peaks in a pH-dependent manner. Cells were stained with 2 μM Lysosensor solution for 5 min, trypsinized, and washed in PBS. The fluorescence of Lysosensor was measured using a Fluorometer (Spectramax M2e, Molecular Devices, San Jose, CA, USA). To determine the relative acidity of the organelles, dual-emitted signals from treated cells were measured in a fluorometer, and the yellow/blue ratio was calculated. For the visual examination of acidic lysosomes, cells treated with the indicated chemicals were visualized by confocal microscopy. To measure lysosomal enzyme activity in situ, cells were prepared as in the experiments of the autophagy flux measurements above. At the end of incubation, 10 μg/mL DQ Green BSA (Thermo Fisher, D12050) was loaded on cells for 30 min. After washing in PBS, cells were examined by flow cytometry. To visualize lysosomes with cathepsin activity, cells were loaded with Magic Red^TM^ (#937, ImmunoChemistry Technologies, Bloomington, CA, USA) according to the manufacturer’s protocol; after fixing, fluorescent puncta were visualized by confocal microscopy.

### 2.8. Statistical Analysis

In all panels, quantification was performed using two or three independent measurements of samples from two different experiments. Data are shown as mean ± standard error (S.D.). Homogeneity of variances was assessed by a Bartlett test, and intergroup comparison of the mean values was performed by one-way analysis of ANOVA using InStat 3.06 (GraphPad Software Inc., San Diego, CA, USA). A *p*-value less than 0.05 was considered to indicate statistical significance.

## 3. Results

### 3.1. Autophagic Flux Impairment in Human Fibroblasts Under Glucose Deprivation

Glucose deprivation has been suggested to induce autophagy, although its flux is attenuated in certain cells [21,41]. We examined the status of autophagic flux in human fibroblasts under glucose deprivation and confirmed these findings. Puncta that are positive for LC3 signal increased as early as 2 h into glucose deprivation and continued increasing at least for 72 h (Figure 1A). Furthermore, multiple results indicated an attenuation of autophagy flux in these cells. The levels of p62 sequestosome (p62), an autophagosome cargo, the decrease of which is a marker for active autophagy flux [42], gradually increased (Figure 1B). Blockade of autophagy flux is also demonstrated in an experiment with bafilomycin A1 (bafA1), which inhibits autophagy flux through the inhibition of autophagosome-lysosome fusion, as well as lysosome acidification [26]. In glucose-deprived cells, the levels of LC3 type II molecules (LC3-II) increased steadily over 72 h, and bafA1 treatment did not further elevate the levels at each time point (shown in Figure 1C, lanes 4 vs. 5 and 7 vs. 8 and 10 vs. 11; this is better demonstrated in the bar graph below, which shows a substantial decrease of the ratio of LC-II proteins in bafA1-treated and untreated cells upon glucose deprivation). This trend is also seen in the case of chloroquine (CQ) treatment, although, in this case, the decrease in the ratio of LC-II proteins upon glucose deprivation. LC-II proteins levels were highly elevated upon chloroquine (CQ) treatment. Chloroquine blocks autophagosome-lysosome fusion but not lysosome acidification [35] and may affect the flux differently from bafA1. Autophagic flux blockade was further indicated in cells expressing a tandem fluorescence-tagged LC3 (tfLC3, a hybrid protein composed of GFP and RFP fused to LC3 [40]) (Figure 1D). In the transfected cells, green puncta represent autophagosomes, while red puncta represent mostly lysosome-fused autophagosomes, where the green fluorescence of tfLC3 is quenched in the acidic conditions of lysosomes [43]. Yellow puncta, which are largely absent in control cells, normally indicate naïve autophagosomes but also can represent autolysosomes with high pH. In cells treated with torin1, which induces ongoing autophagy, the number of yellow puncta increased, which represent autophagosomes. However, in glucose-deprived cells, yellow puncta were more prominent. Together, these data demonstrate impaired autophagy flux in fibroblasts under glucose deprivation.

We also found that co-localization of autophagosomes and lysosomes increased, indicating an accumulation of autolysosomes in glucose-deprived cells (Appendix A). Previous studies reported that cells with autophagic flux problems have irregular and large autolysosomes [44,45]. As shown in Appendix A, both small and large-sized autolysosomes were present among the co-localized structures, and quantitative analysis showed that the numbers of large autolysosomes increased (Appendix A). This pattern is also notable in Figure 1D, as shown by large yellow puncta in the panels of Glu(-) 24 h and 72 h. This increase of numbers of the large puncta also suggests that the flux blockade in glucose-deprived cells occurs at the step of autolysosomal degradation.

### 3.2. Lysosomes Accumulate but Many Are Nonfunctional in Glucose-Deprived Cells

We investigated the molecular mechanisms underlying the failure of autolysosomal degradation in glucose-deprived cells. In glucose-deprived cells, a substantial increase in the number of lysosomes was detected by the increase of Lamp1-positive organelles (Figure 2A and Appendix A) and cellular content of the LysoTracker Red signal (Figure 2B). The increase of Lamp1 and Lamp2 protein levels further supports this finding (Appendix A). This change is attributed at least partially to an increase in lysosome biosynthesis, as evidenced by a gradual increase in the mRNA levels of Lamp1 and Lamp2, as well as cathepsins B and D (Appendix A). To determine whether the increased numbers of lysosomes include lysosomes with defects in degrading cargo materials, we analyzed the activities of the lysosomes using MagicRed or DQ^TM^-BSA, which allow an in situ assessment of the proteolytic activities in endosomes and lysosomes [46,47,48]. Increases in the number of puncta positive for lysosomal protease activities were detected (Figure 2C,D). However, these changes were not as notable as that of the cellular lysosomal content (less than 1.4- and 1.9-fold increases vs. near 4- and 4.5-fold increases at 48 h and 72 h, respectively) (Figure 2B,D). Indeed, the number of lysosomal puncta that co-localized with those positive for Magic Red did not increase in glucose-deprived cells (Figure 2E,F). These findings indicate that a large number of lysosomes are not functional in glucose-deprived cells and that low lysosomal activity underlies the impaired autolysis. 

### 3.3. Lysosomes in Glucose-deprived Cells Exhibit Poor Acidity 

The large yellow tfLC3 puncta, which emit both green and red fluorescence in glucose-deprived cells (Figure 1D), are likely autolysosomes with low acidity and, therefore, cannot quench green fluorescence [49]. In Figure 3A, white puncta, which are indicative of the yellow puncta co-localized with blue Lamp1-positive puncta, increased in number in glucose-deprived cells. The presence of white puncta in cells treated with concanamycin A, a vATPase inhibitor that causes a blockade at the autolysis step [37], supports the possibility that these are indeed autolysosomes with high pH. At 72 h of glucose deprivation, many purple puncta also appeared, suggesting the presence of functional autolysosomes as well. Low acidity of lysosomes in glucose-deprived cells could be determined using a lysosensor yellow/blue dye that emits different fluorescence depending on pH. As shown in Figure 3B, lysosomal acidity decreased in glucose-deprived cells similar as in cells treated with chloroquine or concanamycin A. Decreased lysosomal acidity in glucose-deprived cells was also detected by confocal imaging, as shown in Figure 3C. Glucose-deprived cells showed highly abundant puncta of yellow fluorescence, which also appear in concanamycin A-treated cells but not in control cells, in which fluorescence emission was mostly green. This suggests a dramatic increase in the quantity of lysosomes (or endosomal compartments), most of which harbor low acidity. These results demonstrate that lysosomal acidity is impaired in glucose-deprived cells and suggest that this might contribute to the blockade of autophagy flux. We also found that the acidity of lysosomes and level of autophagosome quantity were recovered by replenishing glucose in the medium (Appendix A). The number of large puncta also markedly reduced (Appendix A). Overall, these findings indicate that impairments of autolysosomal degradation, as well as autophagic flux, are caused by a reversible failure in lysosome acidification.

### 3.4. Increased Reactive Oxygen Species Level Is also Involved in the Failure of Lysosome Acidification and Flux Blockade 

In glucose-deprived cells, the ROS level increases through enhanced OXPHOS [11,12,17,50] (as shown in Appendix A). High levels of ROS have been linked to autophagy flux impairment in cells under certain conditions [51,52,53]. We checked if the increased levels of ROS are responsible for the flux impairment in glucose-deprived cells. In cells undergoing glucose starvation, treatment with *N*-acetylcystein (NAC), a potent antioxidant [20], restored autophagy flux, as evidenced by lowered levels of LC3-II (Figure 4A, lanes 2 vs. 4). A change in flux was also assessed using Cyto-ID dye, which selectively labels functional autophagic vacuoles, including autolysosomes [54]. The level of relative autophagy flux, calculated as the ratio of fluorescent signals in the control, and glucose-deprived cells [55] partially but detectably increased in response to NAC (Figure 4B). A decrease in yellow puncta indicative of the overlap between LC3-positive puncta and lysosomes (Figure 4C) also indicates attenuation in the autophagy flux blockade upon NAC treatment. This treatment also restored lysosomal acidity at least partially but significantly (Figure 4D,E), supporting the possible modulation of lysosome acidity by ROS. The level of lysosomal acidity restored by the NAC treatment was less apparent by fluorometric quantitation compared with in confocal images. The small increase is likely derived from a bias in fluorometric measurement of the change in total cellular acidity. In glucose-deprived cells, lysosomes might lose acidity, but their numbers increase, compensating for the decrease in the total acidity signal in cells. In cells treated with NAC, the populations of lysosomes with high acidity increased, but the total lysosome content decreased (as shown in Figure 4C, lower panel Lamp1 image, and Appendix A). Together, these results indicate that the autophagy flux impairment upon glucose deprivation is caused in large part by increased ROS.

### 3.5. Autophagic Flux Impairment Is Partially Relieved by Inhibition of ATM

The ATM serine/threonine kinase, normally activated by DNA double-strand breaks [28], is also activated by mitochondria-generated ROS [29]. Active ATM has been shown to downregulate the activity of vATPase through direct phosphorylation [56]. The inhibition of ATM activation was shown to help senescent cells maintain vATPase activity, as well as lysosome function [30]. These suggest that high levels of ROS produced from mitochondria in glucose-deprived cells may trigger the activation of ATM and thereby cause the impairment of vATPase activity and lysosome acidity. To check this possibility, we first determined whether ATM inhibition attenuates the impairment of lysosomal acidity and resolves the flux block in glucose-deprived cells. Treatment with KU-60019, a potent ATM inhibitor [30], attenuated the decrease of lysosomal acidity, as demonstrated by confocal imaging of acidic lysosomes (Figure 5A). Fluorometric quantitation also showed a restoration of lysosome acidity, albeit to small extents (Figure 5B). Importantly, KU-60019 reduced the number and size of LC3-positive puncta (Figure 5C) and, also, attenuated the increase in the levels of the LC3-II protein (Figure 5D,E (compare lanes 3 vs. 7 and 5 vs. 9). Additionally, the effect of KU-60019 on the relative levels of bafA1-induced changes in the LC3-II protein (Figure 5E, compare lanes 3 vs. 4 and lanes 7 vs. 8 for day 2 and lanes 5 vs. 6 and lanes 9 vs. 10 for day 3) shows a significant release of the flux block by the treatment of KU-60019. The flux assessment using Cyto-ID also indicated a partial but apparent increase in response to KU-60019 (Figure 5E). Together, these results indicate that a flow in autolysis blocked in glucose-deprived cells is at least partially restored by an inhibition of ATM. Inhibition of ATM also caused a decrease in the levels of lysosomal proteins (Appendix A). This might be an outcome of the removal of autolysosomes, again suggesting a restoration of autolysis upon ATM inhibition. The attenuation of lysosome biogenesis, however, may be another possibility (Appendix A). 

Finally, the small increase in lysosomal acidity and autophagy flux in flow cytometric quantitation (Figure 5B,E) leaves room for a possibility that lysosome acidity and autophagy flux might not be fully restored by the KU-60019 treatment. Therefore, we reasoned there might be other routes of the impairment of lysosome acidity. 

### 3.6. Erk Activation Is Involved in the Impairment of Lysosome Acidity

Previous studies showed that extracellular signal-regulated kinase (Erk) is activated in glucose-deprived cancer cells [10]. Indeed, upon glucose deprivation in fibroblasts, an increase in the level of phosphorylated (active) Erk1 and Erk2 proteins was detected and sustained for over three days (Figure 6A). 

A treatment with NAC or 2,2,6,6-tetramethylpiperidin-1-oxyl (tempol), a superoxide scavenger [57], blocked Erk activation (Figure 6A,B). This demonstrates that ROS generated upon glucose deprivation induced the activation of the MAPK pathway. Importantly, the attenuation of Erk activation by antioxidant treatment alleviated the flux impairment in glucose-deprived cells, as shown by the decrease in the levels of LC3-II in cells treated with tempol (Figure 6B, lanes 3 and 4 vs. 5 and 6). The direct inhibition of Erk activity by PD325901, an inhibitor of MAPK signaling, also lowered the levels of LC3-II (Figure 6C,D). Tempol treatment did not affect the changes in the levels of LC3-II in cells treated with torin1 and undergoing autophagy with normal flux (Figure 6E), suggesting that the change in glucose-deprived cells is an event specific to high levels of ROS. Finally, the treatment of PD325901 during glucose deprivation attenuated the decrease of lysosensor signals (Figure 6F), demonstrating that Erk inactivation prevents or lessens the failure in lysosome acidity. These results together suggest that ROS-induced Erk activation causes poor acidification of lysosomes and impairs autolysis in glucose-deprived cells. We also found that acidic lysosomes in PD325901-treated cells increased but showed fewer numbers of puncta (Figure 6F, right panel) and reduced intensity than those in tempol-treated cells, leaving room for an additional effect of lowered ROS levels in glucose-deprived cells. This may be attributed to the effects contributed by the ATM-mediated impairment of lysosome acidity. However, whether the simultaneous inactivation of ATM and Erk additively increases lysosomal acidity to the levels of NAC or tempol was not determined. 

## 4. Discussion

Our study confirmed an impairment of autophagy flux in glucose-deprived cells and provided further understanding on candidate mechanisms. Here, we show that the flux blockade in glucose-deprived fibroblasts is attributed to impaired autolysis imposed by poor lysosome acidification. At least two factors are involved in this lysosomal failure, ATM and Erk, and both appear to be activated by ROS, which are generated at high levels in glucose-deprived cells. The restoration of lysosome acidity and autophagy flux by the treatment either of antioxidants, the ATM inhibitor or the Erk inhibitor, strongly supports roles of the pathways of ROS-ATM and ROS-Erk. The observation that flux of the autophagy induced by torin1 treatment was not affected by the antioxidant treatment supports the upstream triggering role of ROS in the glucose deprivation-induced impairment of autophagy flux. The involvement of ROS-ATM is easily conceived. ATM is activated by ROS [29] and has also been shown to directly phosphorylate the ATP6V1G1 subunit and inactivate vATPase [30]. Erk activation by ROS generated from mitochondria has also been observed [32,33]. The underlying mechanisms for ROS-induced activation of ATM and Erk upon glucose deprivation deserve further study, though. Little is known for the mechanism underlying the effect of the ROS-Erk pathway on lysosome acidity, although some studies suggest remote clues. A direct interaction of phosphorylated Erk and vATPase was seen in cells with a rotavirus infection [58], but this interaction has not been demonstrated in normal cells. Meanwhile, ERK activation has been shown to attenuate autolysis through a decrease in the cathepsin protein levels [34]. In addition, some reports proposed that active Erk downregulates lysosome biogenesis [59,60]. However, our results of increased mRNA levels of lysosomal genes rule out these possibilities. Another study suggested that, in a condition of mild DNA damage, ATM interacts with and activates Erk [61], suggesting the possibility that Erk activation and vATPase inactivation may also be caused by ATM. Meanwhile, AMPK activation has been proposed to prevent lysosome acidification in mouse embryonic fibroblasts undergoing amino acid starvation [41], suggesting a possible involvement of AMPK activity in the autophagy flux impairment in glucose-deprived cells. In fact, in our study, NAC treatment decreased the level of phosphorylated AMPK (Appendix A). Thus, the impaired lysosomal acidity in glucose-deprived cells might be caused by AMPK activation. However, knocking down the level (albeit, to limited extent) of AMPK by siRNA showed little effect on the lowered lysosome acidity in glucose-deprived cells (Appendix A), suggesting a weak association of AMPK activity with the glucose deprivation-induced lysosome acidity impairment, at least in the tested fibroblasts. Activated AMPK facilitates catabolism to recycle energy resources by activating both autophagy (through mTOR inactivation [62,63,64,65]) and lysosome biogenesis (through the enhancement of the nuclear localization of transcription factor EB (TFEB), a master regulator of lysosomal gene expression [66]). The elevation of autophagosome formation and lysosome biogenesis indeed take place in glucose-deprived cells, indicating that ROS generated upon the hyper-activation of OXPHOS successfully activate AMPK to drive the autophagic recycling of energy resources. However, it appears in this study that the simultaneous activation of ATM and Erk imposes a break to this effort through inactivating vATPase and thereby compromises autophagic recycling. Furthermore, vATPase inactivation appears to override the increased biogenesis of lysosomes. This infertile autophagy activation circuit (summarized in Figure 7) indicates that autophagy activation is not always fruitful as a survival mechanism and can sometimes put cells in unhealthy conditions, such as premature senescence. Accumulated autolysosomes become residual bodies or lipofuscin-containing granules, an increase of which is a marker of cellular senescence and aging [67]. We also observed an increase in lipofuscin-loaded granules in the cytosol of glucose-deprived fibroblasts (SB Song, to be published elsewhere). An accumulation of autolysosomes is also associated with certain diseases and suspected to aggravate certain neurodegenerative diseases by increasing an accumulation of toxic peptides or protein aggregates [68]. Autophagic vacuoles or autolysosomes in the brains of patients with Alzheimer’s disease are proposed to be a major reservoir of Aβ peptide [69,70]. Therefore, preventing or resolving autophagy flux impairment would exert significant effects on certain neural diseases and such conditions. Although it needs to be determined whether high-level ROS are direct triggers of the pathological accumulation of autophagic vacuoles of the disease cells, our results strongly suggest such a role played by ROS and, also, the possible usefulness of antioxidative treatments in the prevention and curing of these diseases. Therefore, approaches to reduce mitochondrial ROS generation or their accumulation would certainly help cells reduce the burden of the autophagy flux blockade and, also, attenuate the accumulation of autophagic vacuoles and thereby alleviate the neuronal degeneration and dysfunction.

## Figures and Tables

**Figure 1 biomolecules-10-00761-f001:**
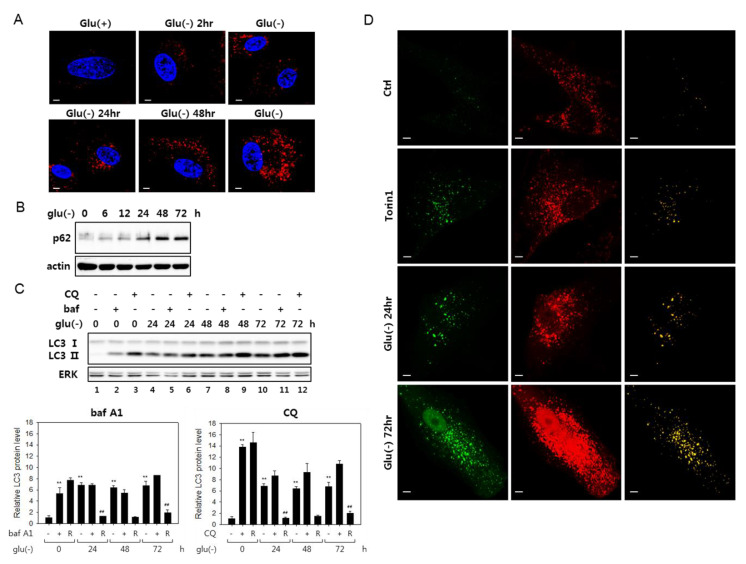
Autophagy flux is impaired in glucose-deprived fibroblasts. (**A**) Human fibroblasts on cover slips were cultivated in normal (Glu(+)) or glucose-free Dulbecco’s modified Eagle’s medium (DMEM) (Glu(-)) for the indicated periods and processed for immunostaining for the LC3 protein (red). Nuclei were stained with 4′,6-diamidino-2-phenylindole (DAPI) (blue). Cells were examined by confocal microscopy. (**B**) Cells cultivated in glucose-free medium were examined by immunoblotting analysis for p62 sequestosome 1 (p62). (**C**) Change in LC3 protein level was determined by immunoblotting in cells cultivated in the absence of glucose for indicated times. The cells were either mock-treated or treated with 200-nM bafilomycin A1 (bafA1) or 50 μM chloroquine (CQ) for 6 h prior to collection for immunoblotting. Relative autophagic flux is shown in the bar graphs (below) of LC-II proteins levels quantitated by densitometry. The LC3-II level in the cells treated with bafA1 or CQ (+) was normalized by that in the cells untreated (-), and the ratio (R) is presented as an indication of the relative flux. (**D**) Cells transfected with plasmid that expresses tandem fluorescence LC3 (tfLC3) were cultivated in the absence of glucose for indicated periods or treated with 250-nM torin1 for 24 h. After fixing, tfLC3 fluorescence was visualized by confocal microscopy. In the right-most column, merged images (yellow) of the green and red fluorescence visualizes structures emitting both green and red fluorescence of GFP and RFP. All scale bars in microscopy indicate 5 μM.

**Figure 2 biomolecules-10-00761-f002:**
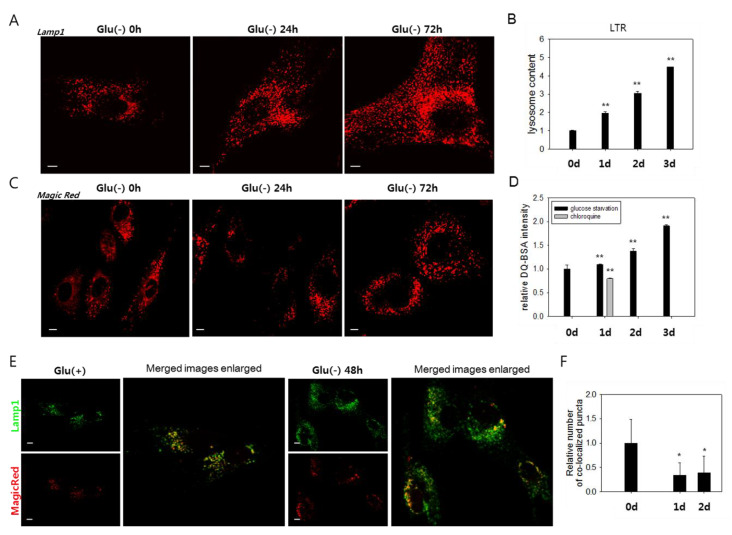
Lysosomes accumulate, but most are poorly functional in glucose-deprived cells. (**A**) and (**C**) Cells were incubated in glucose-free medium (Glu(-)) for 0, 24, or 48 h and then immunostained with anti-Lamp1 antibody (**A**) or incubated with Magic Red dye, a fluorophore-conjugated lysosomal substrate (**C**), and visualized in confocal microscopy. (**B**) and (**D**) More than 2 × 10^4^ cells at each time points were stained with LysoTracker Red (LTR) dye (**B**) or DQ^TM^-BSA green dye (**D**), and the quantitation of fluorescence was performed by flow cytometric analysis. The mean fluorescence signal was determined, and relative values were plotted. In (**D**), cells were glucose-deprived (black bar) or treated with 50 μM chloroquine to generate signals for lysosomes with near-nullified enzyme activity (grey bar). (**E**) Total lysosomes and the ones with enzyme activity were visualized by confocal microscopy. Cells were incubated with anti-Lamp1 antibody (green) or Magic Red (red) and examined by confocal microscopy to detect total lysosomes and enzymatically active lysosomes. The right two panels show overlapping fluorescence images that were enlarged and show puncta co-localization. (**F**) Numbers of co-localized puncta (yellow puncta in (**E**)) in more than 25 cells were counted, and mean values were graphed. All scale bars in photographs indicate 5 μm. Values are presented as the mean of two biological repeats ± S.D. **p* < 0.05 and ***p* < 0.01 by ANOVA.

**Figure 3 biomolecules-10-00761-f003:**
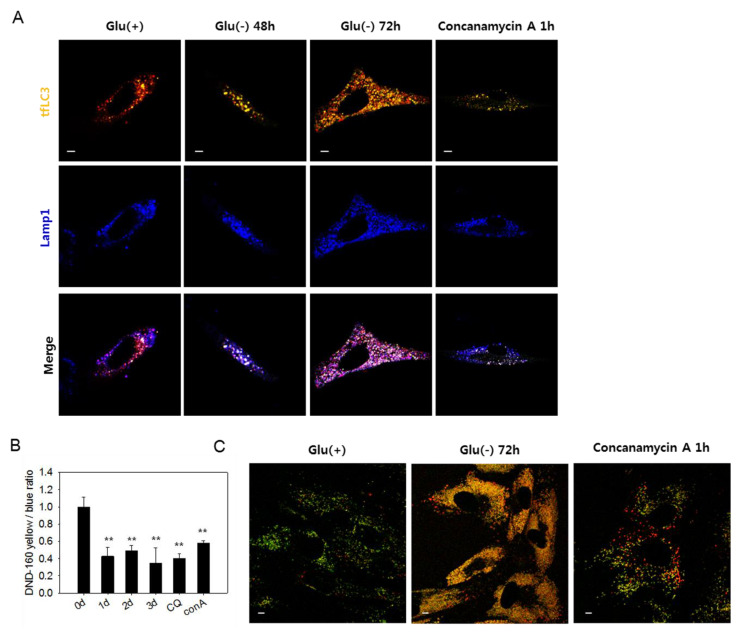
Lysosomes in glucose-deprived cells show low acidity. (**A**) GFP-RFP-LC3 (tfLC3)-positive puncta and lysosomes are visualized by confocal microscopy. Cells were transfected with ptfLC3 plasmid and incubated in the absence of glucose for 48 or 72 h or with 200 nM concanamycin A for 1 h to inhibit lysosome acidification. Cells were fixed, immunostained with anti-Lamp1 antibody (blue), and visualized with confocal microscopy. Merged and enlarged images are presented on the bottom. (**B**) Cells incubated in glucose-free medium for 1, 2, or 3 dor treated with 50 μM chloroquine for 6 h or 200 nM concanamycin A for 1 h were stained with Lysosensor yellow/blue DND-160 dye. After washing in phosphate-buffered saline, cells were examined by fluorometry, and relative lysosome acidity was determined as the relative value of yellow/blue fluorescence ratio. (**C**) Cells were incubated either in the presence or absence of glucose for 3 dor treated with 200 nM concanamycin A for 1 h. Approximately, 2 × 10^4^ cells were then loaded with Lysosensor dye and visualized by confocal microscopy. Vesicles with low and high acidity are seen as red and green puncta, respectively. All scale bars in microscopy indicate 5 μm. Values are presented as the mean of two biological repeats ± S.D. * *p* < 0.05 and ** *p* < 0.01 by ANOVA.

**Figure 4 biomolecules-10-00761-f004:**
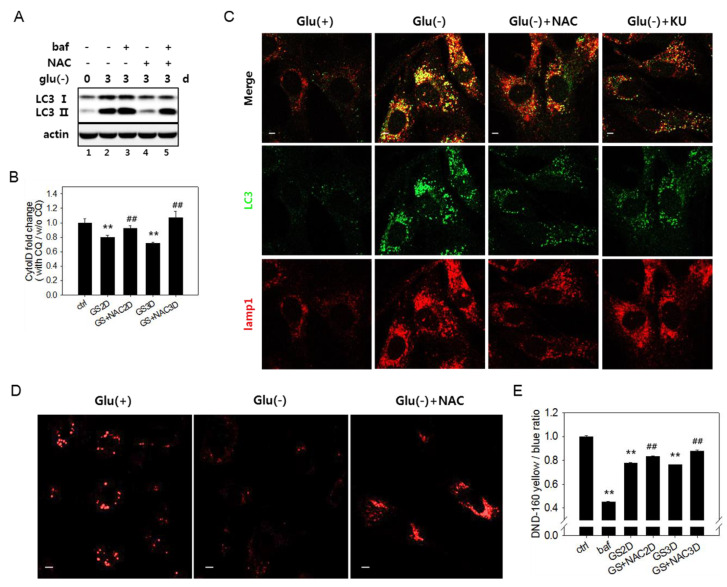
Increase inthe reactive oxygen species (ROS) level is related with a failure of lysosome acidification. (**A**) Autophagy flux was determined by immunoblotting analysis of the LC3-II protein. Cells were glucose deprived for three days in the presence or absence of 5 mM *N*-acetylcystein (NAC). Cells treated with 200 nM bafilomycin A1 (baf) for the last 6 h (lanes 3 and 5) served as a control for a full blockade of flux. (**B**) To determine the relative autophagic flow, cells were treated with 10 μM chloroquine (CQ) during the last 24 h. More than 2 × 10^4^ cells were stained with Cyto-ID dye and examined by flow cytometric quantitation. Relative values (means of two biological repeats) of the measured ratios of fluorescence from CQ-treated and nontreated cells were plotted. # indicates a significance of the difference between the measured means of the cells untreated and treated with NAC. Supplementation of NAC (GS+NAC2D or GS+NAC3D) significantly attenuated the decrease of autophagic flow in cells undergoing glucose deprivation for two or three days (GS2D or GS3D). (**C**) To visualize the co-localization of autophagosomes and lysosomes, cells cultivated with or without 5 mM NAC (Glu(-)+NAC) or 0.5 μM KU60019 (Glu(-)+KU) over an incubation period of three days were subjected to immunofluorescence analysis for LC3 (green) and Lamp1 (red) and observed by confocal microscopy. The upper panels show overlapped and enlarged images. Yellow puncta are abundantly present in glucose-deprived cells but are dramatically reduced in cells treated with NAC. (**D**) Cells on cover slips were cultivated for three days with or without 5 mM NAC, stained with Lysosensor DND-160 dye, and observed by fluorescence of the lysosensor dye emitted at 540 nm. (**E**) Lysosome acidity was determined quantitatively using Lysosensor DND-160 dye. Cells treated with 100 nM bafilomycin A1 for 1 h served as the negative control. All scale bars in microscopy indicate 5 μm. Values are presented as the mean of two biological repeats ± S.D. **p* < 0.05 and ***p* < 0.01 by ANOVA.

**Figure 5 biomolecules-10-00761-f005:**
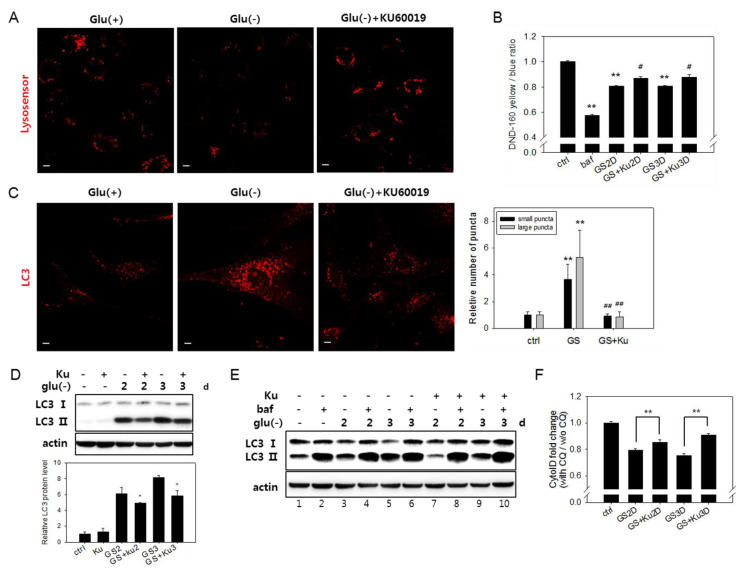
Lysosomal acidity and autophagosome numbers are restored by ataxia telangiectasia mutated (ATM) inhibition. (**A**) Cells on cover slips were incubated with or without 0.5 μM KU60019 for three days. Lysosomal acidity was visualized by treating cells with Lysosensor DND-160 dye and using confocal microscopy with emissions at 540 nm. (**B**) Cells were treated in the absence (GS) or presence of 0.5 μM KU60019 for two or three days (GS+Ku2D or GS+Ku3D). Relative lysosomal acidity was quantitatively determined as the relative value of the yellow/blue ratio of fluorescence measured by fluorometry. # indicates the significance of the difference between the measured means of the cells untreated and treated with KU60019. (**C**) Cells incubated for three days with or without 0.5 μM KU60019 were immunostained with antibody against LC3 and visualized by confocal microscopy. Puncta that are positive for LC3 signal were divided according to diameter (larger or smaller than 1.5 μm), and the numbers of each population in more than 40 cells were counted. Black and grey bars represent the mean numbers of small and large autophagosome puncta in glucose-deprived cells compared with control cells, respectively. (**D**) and (**E**) Levels of the LC3-II protein determined by immunoblotting analysis in cells incubated for two or three days in the absence or presence of 0.5 μM KU60019 (Ku). In (**E**) cells treated with 200 nM bafilomycin A1 (baf) for the last 6 h served as a control for the full blockade of flux. (**F**) To determine the relative autophagic flow, cells were treated with 10 μM chloroquine (CQ) during the last 24 h. Approximately, 2 × 10^4^ cells were stained with Cyto-ID dye and examined by flow cytometric quantitation. Relative values of the measured ratios of fluorescence from CQ-treated and nontreated cells were plotted. # indicates the significance of the difference between the measured means of the cells untreated and treated with KU60019. All values are presented as the mean of two biological repeats ± S.D. * or # *p* < 0.05 and ** or ## *p* < 0.01 by ANOVA.

**Figure 6 biomolecules-10-00761-f006:**
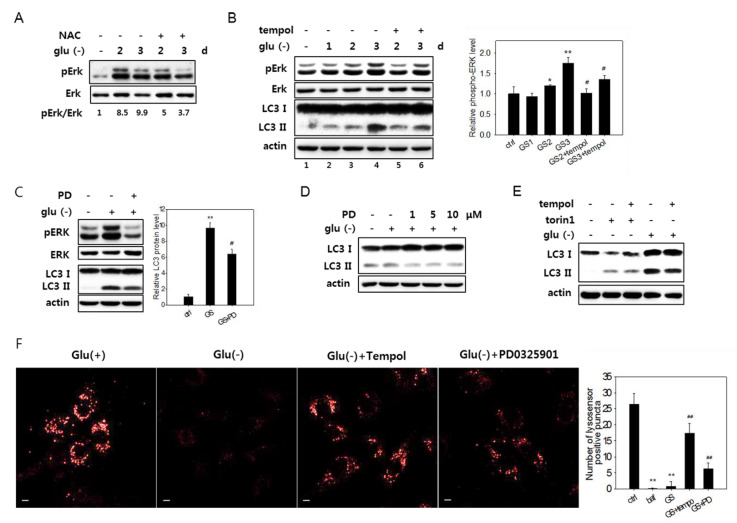
ROS-mediated activation of Erk and autophagy flux impairment in glucose-deprived cells. (**A**) and (**B**) Cells were glucose-deprived (Glu(-)) for the indicated durations in the presence or absence of 5 mM NAC for the last 24 h (in **A**) or 1 mM tempol for the last 6 h (in **B**). Cells were then examined by immunoblotting analysis for Erk, phosphorylated Erk (pErk), and LC3 (only in **B**). Means of two biological repeats of the relative intensities of pErk and Erk bands determined by densitometry are indicated as numbers in A and bars in B. (**C**) and (**D**) Cells were glucose-deprived for three days (in **C**) or one day (in **D**) in the presence or absence of PD0325901 at 1 μM or at 1, 5, or 10 μM for the last 24 h. Cells were then examined by immunoblotting analysis for Erk, phosphorylated Erk (pErk), and LC3 (only in **B**). For the bar graph in C, the mean values of two biological repeats of the relative intensities of the LC3-II band determined by densitometry are plotted. (**E**) Cells were either glucose deprived for three days or treated with 250-nM torin1 for 6 h in the presence or absence of 1 mM tempol for the last 6 h, and the LC3 expression was evaluated by immunoblotting analysis. (**F**) Cells on cover slips were incubated for three days in glucose-free medium with or without the treatment of 1 mM tempol for the last 6 h or 1 μM PD0325901 for the last 24 h. Lysosomal acidity was visualized by treating cells with Lysosensor DND-160 dye, and confocal microscopy was performed using emissions at 540 nm (left). Numbers of positive puncta in more than 30 cells were counted and presented in the bar graph (right).

**Figure 7 biomolecules-10-00761-f007:**
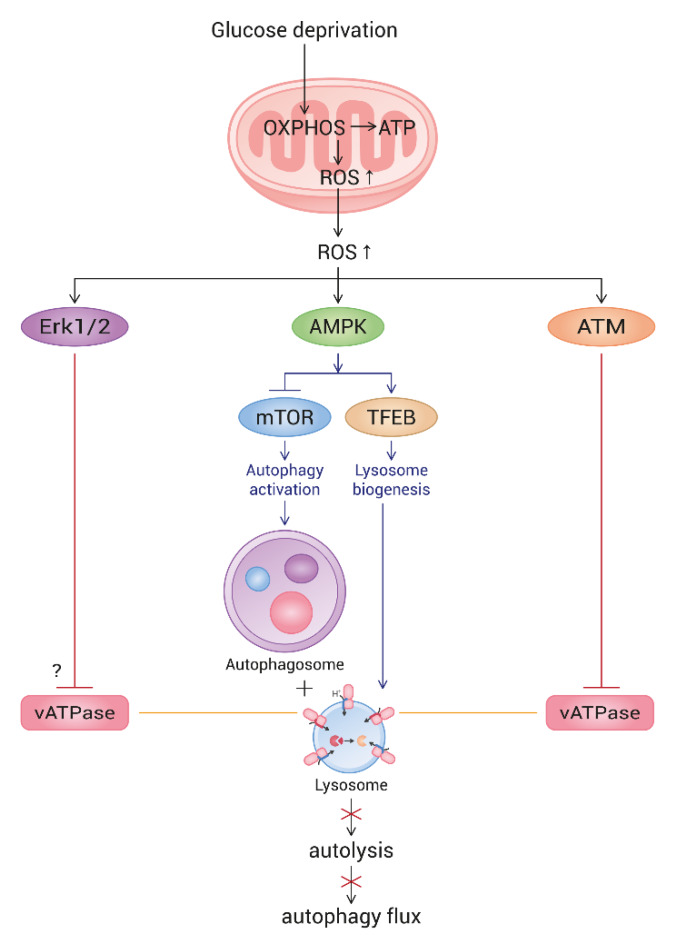
Schematic pathway of ROS-triggered autophagy flux impairment in glucose-deprived cells. Upon glucose deprivation, cells increase mitochondrial ATP production through enhancing OXPHOS. As a result, ROS are highly generated from mitochondria. Mitochondrial elongation also contributes to the elevation of ATP and ROS generation. High levels of ROS activate AMPK, which, in turn, triggers autophagy and lysosome biogenesis. However, ROS also activate ATM and Erk, both of which impair lysosome acidity by the inhibition of vesicular ATPase (vATPase). This overrides lysosome biogenesis and attenuates autolysis. As an outcome, glucose-deprived cells show a blocked autophagy flux. The mechanism underlying the Erk-mediated impairment of lysosome acidity is not yet known.

**Table 1 biomolecules-10-00761-t001:** Chemicals applied to cells.

Name	Manufacturer	Cat #	dose used	Expected effect/property
chloroquine	Sigma-Aldrich, St. Louis, MO, USA	C6628	50 μM	Inhibition of autophagosome-lysosome fusion [35]
bafilomycin A1	Enzo Life Science, Farmingdale, NY, USA	BML-CM110-0100	200 nM	Inhibition of autophagosome-lysosome fusion and lysosome acidification [26]
torin1	Biorbyt, Cambridge, UK	orb146133	250 nM	Induction of autophagy by inhibiting mTOR [36]
concanamycin A	Sigma-Aldrich	C9705	200 nM	Inhibition of vATPase [37]
KU60019	Selleckchem, Houston, TX, USA	S1570	0.5 μM	Inhibition of ATM [30]
n-acetylcysteine	Sigma-Aldrich	A7250	5 mM	Antioxidant [38]
4-hydroxy-2,2,6,6-tetramethyl-piperidin-1-oxyl	Sigma-Aldrich	176141	1 mM	A membrane-permeable free radical scavenger [39]

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
