# Peer review of "High Levels of ROS Impair Lysosomal Acidity and Autophagy Flux in Glucose-Deprived Fibroblasts by Activating ATM and Erk Pathways"

_biomolecules, 2020, doi:10.3390/biom10050761_

Round 1
Reviewer 1 Report
I read the manuscript "High levels of ROS impair lysosomal acidity and autophagy flux in glucose-deprived cells by activating ATM and Erk pathways" with the greatest interest. I think that the authors provided sufficient experimental data for justifying the title of the manuscript. However, I would suggest a small change in the title, by replacing the word "cells" with the word "fibroblasts".
The authors should focus their introduction on the specific phenomena they explored in this manuscript. For example, there is no doubt of the crucial importance of the AMPK function for various metabolic processes in the cell. It should be mentioned in the Introduction, but not in so much detail. I would also suggest that the results obtained on AMPK becomes a regular part of the manuscript rather than just presented on the Suppl. Figure 5. The authors have forgotten to describe some of the roles of ATM, which was explored but given no mention in the Introduction. The same applies to ERK. Thus, the introductory part can be and should be improved in a fashion that would tailor its content to accord with what was presented.
There are some details that should be improved in the MM section:
Is there a consent of the Ethical Committee for using the newborn’s skin in these experiments?
For the seven compounds listed under 2.1., it would be beneficial for readers, including this reviewer, to have prepared a Table listing their specific effects. In the way it is written now, it is not easy to follow the specific application of a particular listed inhibitor.
The authors state that "In re-feeding experiments, 5.5 mM glucose was added to the glucose-free medium". The question is what was the final concentration of glucose in these re-feeding experiments?
In this section, the authors list the anti-TP53 antibodies, but they did not use them, much as I can see.
The Results section:
With respect to Fig. 1C, was there any densitometric analysis performed? If not, then it should be performed. Why was not the housekeeping protein shown on Fig. 1C?
With respect to Suppl. Fig. 2B, it should be clarified what the bars in different colors present. Is it a presentation of the level of the target transcripts during the different time windows of glucose starvation? That is not clear.
Figure 6 should proceeds the Discussion section.
It is a pity that ROS was not measured, although application of N-acetylcysteine clearly pointed out the background of phenomena observed.
The Discussion is also partially dispersed and some parts of it are reminiscent of a review article. The Discussion itself introduces 20 new articles. While I personally like and prefer in depth-discussion, in this manuscript it should be reduced and primarily closely and functionally connected to the experimental data presented.
Thank you.
Reviewer 2 Report
This study examines the outcome of autophagy induction and processing (flux) in response to glucose deprivation in primary fibroblasts isolated from foreskin. The premise for the study is that there is some evidence suggesting that glucose deprivation in this cellular model initiates autophagy (as would be expected) but that the secondary activation of ROS (via OXPHOS and mitochondria dysfunction) prevents the processing of autophagic vesicles (autolysis). Therefore, in this study, the authors investigate the involvement of ROS/DNA damage on the autophagic flux by assessing lysosomal activity and the effect of inhibiting DNA damage response (ATM inhibitor) and ERK activation. The authors conclude that ATM inhibition and ERK inhibition restore lysosome activity (measured through lysosome acidity).
The data in Fig. 1 to 3 is convincing. However, the effects reported in Fig 4, 5 and 6 are marginal, particularly in Fig. 6, panels C and D (see specific points). Unfortunately, this considerably reduces enthusiasm for the findings and makes the conclusions very speculative.
The main issue globally is that these data are generated in one cell type (primary fibroblasts) and one wonders whether the ROS activation is specific for this cell type. The fact that ATM is activated (this is actually not shown in the study) and that ERK is activated is not surprising given that ROS will activate a DNA damage and stress response. The authors suggest that inhibiting either ATM or ERK has an effect on autophagic flux. The results are marginal and there are some experimental issues (see below).
Overall, it appears that the conclusions stated in the abstract are poorly supported by the data. For instance the abstract states: “A large population of lysosomes exhibits low acidity, and this is mainly caused by high levels of ROS”. This is overstated, as figure 4 states that “ increased ROS is related with failure of lysosome acidification”. “Related” is the operative word. There is a correlation, but no causal effect.
Another issue with the abstract is in the statement: “Inhibition of ATM serine/threonine kinase, which is activated by ROS, also attenuated the impairment of lysosomal acidity and autophagic flux, suggesting an effect of ROS mediated by ATM activation”. These experiments are poorly controlled (see below) and the results are quite marginal.
Finally, the concept that there is an impairment of autophagic flux in glucose-deprived cells is puzzling. Autophagy is a survival mechanism that is established to respond to a lack of nutrients, or energy, so why should it be interrupted? What is the meaning of this response? This is particularly confusing since the authors employ primary skin cells that are not pathologically altered. Is it possible that the culture conditions of these primary cells increases oxidative stress which would induce such a response? This is not discussed at all by the authors and should be addressed.
Specific points:
Fig. 4B, Fig. 4E and Fig. 5B and 5E are difficult to interpret as controls are missing. There is only 1 control (ctrl) and then treatments for 2 or 3 days. How does this one control compares to the treated samples? There should be a control for each day of culture (day 2 and day 3 controls). Also, there are no indications of how many samples have been used (n=??). The differences are extremely small and the error bars are also extremely small despite being SD….
Fig. 5D: Effect of ATM inhibitor (Ku-600190 on the autophagic flux is not obvious to this reviewer. The way that the samples are loaded makes it very difficult to evaluate the data. One wants to compare the effect of the ATM inhibitor (Ku-60019) with equivalent untreated cells but the samples are not loaded next to one another so the reader has to compare lanes 3 with 7 ad 5 with 9. These should be next to one another, so samples should be reloaded. But, more importantly, this reviewer cannot see much difference between untreated and +Ku 60019 in lanes 5 and 9….so where is the effect of the ATM inhibitor? Also, the treatment should ideally be done on cells that are not glucose-deprived to evaluate possible non-specific effects of the inhibitor.
Fig. 6: it is postulated in the text of the manuscript that because the effect of ATM inhibition is partial, then other mechanisms may be involved. The authors chose to look at ERK, and they report a small effect of ERK inhibition on autophagy processing. So would both ATM and ERK inhibition result in a larger effect? This should be tested.
In all figures showing data quantifications (Fig 2-6), the n values are missing. Details should also be given as to how many cells were counted (for instance in Fig 2B, D, E or in Fig. 3B, Fig. 5C, 6F etc…)
All abbreviations used in the figure panels should be explained in the figure legends.
Molecular weight markers should be indicated on western blot panels.
Reviewer 3 Report
The work in this manuscript investigates how accumulation of reactive oxygen species (ROS) induced by glucose deprivation can reduce lysosomal acidity and thus affect autphagic flux. The authors further claim that lysosomal dysfunction and autophagy blocking is regulated by ATM and the Erk MAPkinase signalling pathways. The manuscript is interesting and the methods, which were applied are adequate although mainly focusing on fluorescence microscopy and immuno-blotting. The figures need adaptation for clarity (see minor points). The hypotheses of ATM and Erk involvement are only based on using specific inhibitors. An additional approach using siRNA mediated knock down would strengthen the claim. In Fig. S5 siRNA-mediated AMPK knock-down has been performed. However, the efficiency of the knock down is not very convincing and the claim should thus be softened.
The important question of how ROS could possibly activate ATM and Erk pathways remains unaddressed in this manuscript. Are they direct targets and potentially modified by ROS? A biochemical approach to investigate this could be conducted in a follow-up study and/ or be discussed towards the end of the manuscript.
Major point:
- Substantiate ATM and Erk involvement by siRNA-mediated knock down.
Minor points:
- Line 10: remove “of”
- Line 69: for clarity please rephrase: “but instead induced by an increase in ROS”
- Line 73: Citation of own manuscript is missing
- Line 92: The manuscript does not provide support that ROS are exclusively generated by mitochondria. The authors should delete the word “mitochondrial”.
- Cite Kimura et al. 2007 in the methods part as well when describing the ptfLC3 construct.
- More detail for statistical analysis is needed. E.g. How was normal distribution assessed?
Figures:
- Diagrams containing bars in more than one color should have a legend (e.g. Fig. 2D, 5C, S1, S2…)
- Reorder panel labelling in Figure 2, so that they appear sequentially in the main text. (swap B with C).
- Line 251: replace “active” with “activity”
- I suggest repositioning the merged/enlarged images in Fig. 2 E directly next to the small ones for clarity regarding the time point and make the puncta quantification a separate panel (F)
- 3A: Position the small pictures to fit the magnifications. What is the time point of Concanamycin treatment. Introduce abbreviation “GS” for glucose starvation
- 4C: Position the small pictures to fit the magnifications.
- 6A: Can you comment on the two bands observed with the pERK-antibody.
- Line 407: Citation is missing
- Line 473: “mTOR inactivation”
- Figure legends for supplemental figures are missing and need to be provided for understanding of the figures.
Round 2
Reviewer 2 Report
Upon evaluation of the revised version of the manuscript from Song and Hwang, it appears that the authors have revised their manuscript but there are still a number of issues that not been answered adequately. Also, there are a number of additional issues in the revisions that were made by the authors. The most important concern statistical analyses that were done with 2 samples according to the authors (and mentioned in the manuscript. As mentioned below, this small number of samples raises questions regarding the validity of the results.
First, it is not clear to this reviewer exactly what text was revised in this manuscript. The revised text should be indicated in red. However there is red text in the revised manuscript that was present in the original manuscript, therefore was not changed. For instance, the first 2 sentences of the abstract have text colored red, however these 2 sentences are identical to the original manuscript, meaning no changes were made. Similarly, some words are indicated in red in the first sentence of the result section, however, the sentence is identical to the one in the original version. While the text in some large sections has clearly been changed, why has some text been indicated as changed when it has not?
In the abstract, some modifications have resulted in sentences that don’t make sense, perhaps due to poor English syntax: for instance, it is unclear what the following statement means: “ an involvement of high level ROS-induced activation of Erk and ATM in this impairment.”
The authors mentioned in the response to reviewer that (in Figure 6C) “a bar graph that shows the changes in LC3II bands quantitated in densitometry on the blots of two biological repeats is added in the revised submission.” Also, in Figure 5, it is mentioned: “All values are presented as mean of two biological repeats “. How can one make any conclusions with only 2 samples? This is not a valid analysis to this reviewer, and a minimum of 3 samples is generally considered enough for statistical analyses, so these quantifications are meaningless.
The authors still have not indicated the number of samples used for their statistical analyses (n= ) in the figures legends of Figures 4 and 6 , as was requested in the previous review.
The previous review also requested to indicate molecular weight markers on western blot panels. This has not been done.
In addition, the authors mentioned in their rebuttal for experiments shown in Figure 6 that “ due to time restraint for the revision submission, we could not carry out additional experiments.” The authors can and should ask for an extension to have more time to do necessary experiments. That should not be a problem and should be done.
Reviewer 3 Report
The points raised have been addressed adequately.
Author Response
We sent to the reviewer.